# Aflatoxin Biosynthesis and Genetic Regulation: A Review

**DOI:** 10.3390/toxins12030150

**Published:** 2020-02-28

**Authors:** Isaura Caceres, Anthony Al Khoury, Rhoda El Khoury, Sophie Lorber, Isabelle P. Oswald, André El Khoury, Ali Atoui, Olivier Puel, Jean-Denis Bailly

**Affiliations:** 1Toxalim (Research Center in Food Toxicology), Université de Toulouse, INRAE, ENVT, EI-Purpan, 31300 Toulouse, France; isauracaceres@hotmail.com (I.C.); anthony_ek@hotmail.com (A.A.K.); rhodakhoury@gmail.com (R.E.K.); sophie.lorber@inra.fr (S.L.); isabelle.oswald@inra.fr (I.P.O.); olivier.puel@inra.fr (O.P.); 2Centre d’analyse et de recherche, Unité de recherche technologies et valorisations agro-alimentaires, Faculté des Sciences, Université Saint-Joseph, P.O. Box 17-5208, Mar Mikhael Beirut 1104, Lebanon; andre.khoury@usj.edu.lb; 3Laboratory of Microbiology, Department of Life and Earth Sciences, Faculty of Sciences I, Lebanese University, Hadath Campus, P.O. Box 5, Beirut 1104, Lebanon; aatoui@ul.edu.lb

**Keywords:** aflatoxin, biosynthesis, gene regulation, *Aspergillus*

## Abstract

The study of fungal species evolved radically with the development of molecular techniques and produced new evidence to understand specific fungal mechanisms such as the production of toxic secondary metabolites. Taking advantage of these technologies to improve food safety, the molecular study of toxinogenic species can help elucidate the mechanisms underlying toxin production and enable the development of new effective strategies to control fungal toxicity. Numerous studies have been made on genes involved in aflatoxin B1 (AFB1) production, one of the most hazardous carcinogenic toxins for humans and animals. The current review presents the roles of these different genes and their possible impact on AFB1 production. We focus on the toxinogenic strains *Aspergillus flavus* and *A. parasiticus,* primary contaminants and major producers of AFB1 in crops. However, genetic reports on *A. nidulans* are also included because of the capacity of this fungus to produce sterigmatocystin, the penultimate stable metabolite during AFB1 production. The aim of this review is to provide a general overview of the AFB1 enzymatic biosynthesis pathway and its link with the genes belonging to the AFB1 cluster. It also aims to illustrate the role of global environmental factors on aflatoxin production and the recent data that demonstrate an interconnection between genes regulated by these environmental signals and aflatoxin biosynthetic pathway.

## 1. Introduction

Aflatoxins are toxic secondary metabolites produced by several fungal species mostly belonging to *Aspergillus* section *Flavi* [1]. Toxigenic fungi in this section synthesize four main molecules, aflatoxin B1, B2, G1 and G2, all of which can yield other derivatives when metabolized by animals after their ingestion. Aflatoxin B1 (AFB1) is one of the most important compounds due to its demonstrated carcinogenic properties in human and its frequent presence in many foodstuffs worldwide [2]. Studies demonstrated that chronic exposure to AFB1 can lead to numerous diseases including immune suppression in humans and animals, malabsorption of nutrients, infertility, endocrine problems as well as teratogenic effects related with congenital malformations and hepatocellular carcinoma [3]. Moreover, it has been recently demonstrated that intermediates precursors compounds within the aflatoxin biosynthetic pathway, such as versicolorin A, could also represent a potential risk due to their cytotoxic effects [4]. 

It was long assumed that AFB1 contamination was a major public health issue in tropical and subtropical regions because the climate may favor the development of aflatoxigenic species in the field or during storage. However, with ongoing global climate changes, AFB1 is predicted to be an emerging threat in areas where it was not previously present [5,6]. Such is the case of several regions of Europe [7,8]. The genotoxic property of AFB1 justifies limiting consumer exposure to this toxin to the greatest extent possible as reflected in worldwide regulations allowing only a few µg of the toxin per kg of food [9]. To reach such ambitious objectives, many preventive strategies have been developed to reduce AFB1 occurrence in food commodities. These strategies range from good agricultural practices to the use of biocontrol agents or natural compounds able to block toxin production [10]. Even if the exact mechanism of action has not yet been completely elucidated, a few studies already demonstrated that some of these natural products can inhibit AFB1 production by a transcriptional down-regulation of the genes involved in AFB1 synthesis [11,12,13,14,15,16,17,18]. Aflatoxins are the product of a complex biosynthesis pathway involving at least 27 enzymatic reactions [19,20,21,22,23,24]. The genes coding for these enzymes are grouped in a cluster and their expression is coordinated by two cluster-specific regulators: *aflR* and *aflS* [25,26]. Nevertheless, as a secondary metabolite, AF synthesis also depends on other complex mechanisms triggered in response to environmental stimuli including pH, light, nutrient sources and oxidative stress response, which may activate different cell signaling pathways resulting in the modulation of the expression of genes involved in toxin production [27,28,29,30,31]. The recent development of molecular tools in the field of fungal physiology allowed the demonstration of the interaction between many genes involved in response to environmental stimuli and the AF cluster, even if the exact level of connection is often not completely elucidated. Understanding the connection between the AF cluster and environmental stimuli may help to define new strategies to limit mycotoxin production by specifically targeting genes involved upstream of the cluster of toxins.

As one of the most hazardous mycotoxins, there are many reviews concerning different aspects of aflatoxin biosynthesis [29,32,33,34,35,36,37]. Thus, the present review is timely and provides an overview of the different factors interacting with the AFB1 gene cluster and hence with toxin production. However, an effort to link these factors in order to synthesize the global network regulating aflatoxin production is proposed. Data on *Aspergillus flavus* and *A. parasiticus*, the two main producers of aflatoxins in crops, are presented. Studies of *A. nidulans* linked with sterigmatocystin production are included since this mycotoxin is the penultimate stable intermediate in the AFB1 synthesis cascade and its cluster shares 25 homologous genes with the AF cluster [24]. After reminding the AFB1 biosynthetic pathway and its internal gene regulation, we describe other genes found to interact with AFB1 synthesis according to their role in fungal metabolism. These genes are divided into several categories: cell signaling, reproductive process, growing conditions and oxidative stress as shown in Figure 1. 

## 2. Aflatoxin Biosynthetic Pathway

### 2.1. The Aflatoxin Gene Cluster

In *Aspergilli*, DNA information is organized in 8 chromosomes where genes responsible for aflatoxin production are located in the 54th cluster, 80 kb from the telomere of chromosome 3 [29]. This cluster includes 30 genes and its activation is mainly regulated by *aflR* and *aflS* [25,26] (Figure 2). 

The aflatoxin gene cluster has been widely studied in *A. flavus* and *A. parasiticus*. The homology of the clustered genes between the two fungal species is 90–99% [38]. One of the main differences between the two species is their ability to produce B and G type aflatoxins. *A. flavus* mainly produces B type AFs (AFB1 and AFB2) whereas *A. parasiticus* produces B and G type molecules (AF-B1, -B2, -G1 and -G2). Functional genes involved in G-type aflatoxin production correspond to *aflU, aflF* and *nadA* respectively coding for a cytochrome P-450 monooxygenase, an aryl alcohol dehydrogenase and an oxidase [39,40]. Understanding the AF cluster has also benefited from experiments on *A. nidulans* regarding its capacity to produce sterigmatocystin. Indeed, the homology between *A. parasiticus* and *A. nidulans* clusters is 55–75% [38].

### 2.2. Enzymatic Cascade Leading to Aflatoxin Synthesis

Aflatoxins are produced by a polyketide pathway that was first proposed by Birch in 1967 [41] and today, at least 27 enzymatic reactions have been shown to be involved in this process. It has long been considered that, among natural secondary metabolites, aflatoxin biosynthesis is one of the longest and most complex processes due to the quantity of oxidative rearrangements it includes [42]. Three critical oxygen elements were characterized in this pathway by Dutton in 1988 [43]:(i)Monooxygenases: responsible for incorporating one oxygen atom in another being reduced, with nicotinamide adenine dinucleotide phosphate (NADPH) acting as a cofactor.(ii)Dioxygenases: involved in ring-cleavage reactions. (iii)Baeyer-Villiger reactions: responsible of inserting oxygen atoms between two carbons. 

Cytochromes P-450 also play a key role in aflatoxin synthesis. These enzymes are involved in attaching functional groups (i.e., methyl, acetyl) during biosynthesis [44]. Indeed, aflatoxin gene cluster contains the highest number of cytochromes P-450 among the mycotoxin biosynthetic pathways known to date [45].

In order to schematize the principal enzymatic reactions occurring during AFB1 production, Figure 3 shows the various intermediates in the biosynthesis of AFB1 as well as the genes present in the aflatoxin gene cluster responsible for encoding the enzymes for their synthesis. 

To remind the specific characteristics and interventions of the genes participating in the AFB1 biosynthesis pathway, it will be presented in four paragraphs according to the stable metabolites that appear during AFB1 production: norsoloric acid, versicolorin A, sterigmatocystin and finally, aflatoxin B1. Genes are indicated with new and former (in brackets) nomenclatures.

### 2.3. Conversion of Acetate into Norsolorinic Acid

Aflatoxins are polyketide-derivates requiring the formation of hexanoate units (from acetyl-CoA and malonyl-CoA) to start the cascade reaction [23]. The first steps in the pathway lead to the transformation of the starter unit into the first stable metabolite norsolorinic acid (NOR). The enzymes involved in these reactions are mainly encoded by four genes: 

- *aflA* (*fas-2*) and *aflB* (*fas-1*) formerly named “*fas”* genes as they code for fatty acid synthases. Their corresponding synthesized proteins are α and β sub-units primarily responsible for transforming the hexanoate units into a polyketide structure [23,56]. 

- *aflC* (*pksA*) is a gene coding for the synthesis of polyketide skeletons. In general, secondary metabolites that are acetate derivatives, as is the case of Aflatoxin B1, are subject to chain elongation induced by this polyketide synthase (e.g., from 2 MalonylCoA to 7 MalonylCoA). In addition, this enzyme is involved in further transformations of the polyketide structure into norsolorinic acid anthrone (NAA) [23]. 

- *hypC* (*hypB1*) is a gene located between *aflC* and *aflD*. It encodes for a 17-kDa enzyme that was demonstrated to be involved in the catalytic conversion of NAA into NOR [57]. 

### 2.4. Conversion of Norsolorinic Acid into Versicolorin A 

Norsolorinic acid is further transformed into averantin (AVN), in a step governed by the *aflD* gene. For years, the involvement of two other genes (*aflE* and *aflF*) was associated with this step. However, further studies provided evidence for their implication in other steps in the AFB1 enzymatic cascade, and their role is described later.

*- aflD* (*nor-1*) encodes a norsolorinic acid ketoreductase needed for the conversion of the 1′-keto group of NOR to the 1′-hydroxyl group of AVN [58]. In the 1990s, Yabe et al. [59] identified, in *A. parasiticus*, two enzymatic reactions involved in the conversion of averantin into averufin (AVF). This conversion first involves a P-450 monooxygenase responsible for converting the AVN into 5′-hydroxyaverantin (HAVN) and second, the transformation of HAVN into averufin by an alcohol dehydrogenase. The transformation from HAVN into AVF was originally thought to occur in one step. However, Sakuno et al. [50] discovered that an intermediate compound identified as 5′-oxoaverantin (OAVN) is generated during the process. The genes responsible for the above enzymatic processes are: 

- *aflG* (*avnA*), encoding a cytochrome P-450 monooxygenase that catalyzes the hydroxylation of the polyketide anthraquinone averantin into 5′ -hydroxyaverantin [56,60,61]. 

- *aflH* (*adhA*), coding for the alcohol dehydrogenase that is needed for the conversion of HAVN into 5′-oxoaverantin (OAVN) [62]. 

*- aflK* (*vbs*) was firstly associated with the conversion of versiconal into versicolorin B but was also reported to be responsible for the transformation of OAVN into AVF [47]. Indeed, when Sakuno et al. [47] elucidated the intermediate metabolite between HAVN and AVF, the 5′-oxoaverantin (OAVN), this transformation was also associated with the *aflK* gene. These authors were the first to demonstrate that the same enzyme can catalyze two different reactions in the AFB1 pathway. They hypothesized that this phenomenon is due to the evolution of the AFB1 gene cluster that may previously have comprised two copies of the *aflK* gene. 

Further conversion from averufin into versicolorin B is governed by different genes: 

- the *aflI* (*avfA*) gene is involved in the transformation of averufin (AVF) into versiconal hemiacetal acetate (VHA). Indeed, the deletion of this gene led to the accumulation of AVF [38] and it is generally assumed that the enzyme encoded by *aflI* catalyzes the ring-closure step during the formation of hydroxyversicolorone (HVN). Nevertheless, the exact role of AflI in the oxidation of AVF has never been elucidated [24]. 

- *aflV (cypX)* and *aflW* (*moxY*) encode a P-450 microsomal monooxygenase and a cytosolic monooxygenase, respectively [48]. AflV has been shown to catalyze the reaction from AVF to HVN and AflW to catalyze the transformation of HVN to VHA by a Baeyer–Villiger reaction. 

The further transformation from VHA into versiconal (VAL) is governed by the gene *aflJ*, after which *aflK* is responsible for the transformation of VAL into versicolorin B (VERB). 

- *aflJ (estA)* encodes an esterase catalyzing the conversion of VHA to versiconal (VAL). Interestingly, this enzyme has been reported to allow the reversible transformation of VHA into versiconol acetate (VOAc) and from versiconol (VOH) to VAL, *aflJ* being therefore responsible for both reactions during AFB1 biosynthesis [51,63]. 

- *aflK* (*vbs*) encodes a cyclase involved in the transformation of VAL into VERB [64] but also, as already mentioned, during the transformation of OAVN into AVF. The coded enzyme is responsible for bisfuran ring closure in aflatoxins, which, in turn, is responsible for DNA binding after metabolization and the mutagenic effect of aflatoxins [33]. 

Finally, in *A. nidulans*, the transformation of VERB into versicolorin A (VERA) is presumed to be done by the product of the *aflL* gene, coding for a cytochrome P-450 monooxygenase/desaturase [65]. The main characteristic of this step refers to the final metabolic transformation before the principal branch leading to the synthesis of B or G-type aflatoxins [46]. 

### 2.5. Conversion of Versicolorin A into Sterigmatocystin

Four genes are involved in the transformation of VERA into demethylsterigmatocystin (DMST):

- *aflM* (*ver-1*), predicted to encode a ketoreductase, is involved in the conversion of VERA into DMST [52] and its homologous gene in *A. nidulans* has been also identified [66]. Further studies demonstrated that the promoter of this gene contains cAMP-response element sites (CRE) that are also present in genes involved in fungal oxidative stress response [67]. However, the exact intervention of this enzyme has not been yet completely elucidated. 

- *aflN* (*verA*) codes for a cytochrome P450-type monooxygenase which exact function also remains to be determined. Some studies suggest that this protein could be involved in the passage of VERA into a hypothetical intermediate [52]. Its homolog in *A. nidulans* corresponds to *stcS* [68]. 

- *aflY* (*hypA*) seems to mostly intervene between two hypothetical intermediate structures between the transformation of VERA into DMST by a Baeyer–Villiger reaction. Disrupted *A. parasiticus* strain Δ*aflY* accumulated VERA, which is the main reason to include this gene in the VERB-DMST group [54]. 

- *aflX* (*ordB*) encodes an oxidoreductase that catalyzes the oxidative decarboxylation and ring-closure of the Baeyer–Villiger intermediate resulting from the AflY-catalyzed oxidation [53]. 

- *aflO* (*omtB*), which codes for an *O*-methyltransferase, is involved in the conversion of DMST into sterigmatocystin (ST), the penultimate intermediate in the AFB1 pathway and the final metabolite in *A. nidulans* [38]. 

### 2.6. Conversion of Sterigmatocystin into Aflatoxin B1

*- aflP* (*omtA*) codes for one of the enzymes identified in *A. parasiticus* as an *O*-methyltransferase. It is one of the main genes responsible for transforming ST into *O*-methylsterigmatocystin (OMST) [69]. This gene has been shown to be expressed only in aflatoxin permissive conditions [70]. 

The final conversion of OMST into AFB1 is governed by the following genes: *aflQ*, *hypB*, *aflE* and *hypE*. During this final transformation, the intervention of *aflQ* and *hypB* has been precisely identified while only partial information is reported for the other genes. 

- *aflQ* (*ordA*) an adjacent gene to *aflP* in the AFB1 cluster codes for a P-450 monooxygenase. This gene is involved in the conversion of OMST into AFB1 by the oxidation of the A-ring of OMST [24,71]. This reaction leads to an AFB1 precursor named 11-hydroxy-*O*-methylsterigmatocystin (HOMST) [55]. 

- *hypB* (*hypB2*) is a gene coding for an oxidase involved in the second transformation step from HOMST into a 370 Da 7-ring lactone and expressed under aflatoxin permissive conditions [57]. The transformation of this compound into another unknown intermediate is hypothesized to be catalyzed by hydrolytic enzymes encoded by genes that do not belong to the AF cluster [24]. 

- *aflE* (*norA*) is a homologous gene of *aflD* in the aflatoxin cluster encoding a short-chain aryl alcohol dehydrogenase [22]. The first characterization of *aflE* in *A. parasiticus* suggested that this gene was involved in the transformation of NOR to AVN [72]. However, further studies by Ehrlich et al. [73] in *A. flavus,* demonstrated that *aflE* was mainly involved in the two final steps of AFB1 formation (even if the exact position has not yet been identified) since the absence of *AflE* led to the accumulation of deoxyaflatoxin.

- *hypE* (*aflLa*) is a gene responsible for the final steps of AFB1 production since its disruption leads to an intermediate compound before deoxyAFB1 synthesis. HypE presents homologies with several bacterial enzymes due to its ethD domain and it was thus suggested that, together with a P-450 monooxygenase, it probably interacts in the aflatoxin enzymatic pathway [24].

As demonstrated, AFB1 production is a long and complex process corresponding to a coordinated enzymatic cascade in which at least 27 enzymes coded by genes belonging to the AFB1 cluster are involved. Nevertheless, to date, the roles of several other genes, also belonging to the cluster, remain unclear. 

## 3. Genes Present in the AFB1 Cluster with an Unclear Role in the Aflatoxin Enzymatic Cascade

Within the AFB1 gene cluster, two genes have not been directly associated with the biosynthetic pathway:

*- aflT* is a gene coding for a fungal transporter that belongs to the major facilitator superfamily (MFS). This gene was widely characterized by Chang et al. [74] who demonstrated that even if it resides in the AF gene cluster, its role is not directly linked to aflatoxin biosynthesis. In fact, its deletion in *A. parasiticus* did not affect the final amounts of aflatoxins compared to the control. These results are in agreement with others reporting that *aflT* is neither regulated by the main activator of the AFB1 pathway *aflR,* nor by *aflS* but instead by an external factor, FadA, a sub-unit of the G-protein signaling pathway [74]. In the same study, experiments using *S. cerevisiae* yeast led the authors to suggest that *aflT* is not implicated in the transport of aflatoxin. However, Chanda et al. [75] reported that AflT resides in the aflatoxisomes, which are implicated in the exocytose of aflatoxins. So, the exact role of *aflT* in relation with aflatoxin production is still not clear. 

*- hypD* (*aflNa*) coding for a 129 Da integral membrane-binding protein is another gene whose role is still not clear. Ehrlich [24] reported that deletion of *hypD* in an *A. parasiticus* strain resulted in increased sporulation of cultures along with reduced levels of AFB1 production, suggesting that HypD is involved in both fungal development and toxin production [15,18]. 

## 4. Aflatoxin Cluster-Specific Regulators

### 4.1. The aflR Transcription Factor

In *A. flavus*, *A. parasiticus* and *A. nidulans,* the aflatoxin and sterigmatocystin biosynthetic pathways are mainly regulated by the *aflR* gene [31,76]. In *A. flavus*, the AflR protein binds to at least 17 of the genes in the AF cluster that results in the activation of the enzymatic cascade and leads to the production of different AFs. AflR is classified as a zinc cluster Zn(II)_2_Cys_6_ transcription factor of the Gal4-type family [77]. This type of transcription factor has a specific structure that is only attributed to the fungi kingdom. Such transcription factors are able to bind to DNA using a DNA-binding domain, which is one of the most important elements in transcriptional and translational processes [78]. In fact, the regulation process occurs in the nucleus. Once there, the zinc finger parts of AflR bind to DNA and it is believed that this binding occurs in a homodimer manner [78,79]. In particular, AflR was demonstrated to preferentially recognize the palindromic pattern 5′-TCG(N5)CGA-3′, but it also binds to other sequences such as 5‘-TTAGGCCTAA-3′ and to the 5′-TCGCAGCCCGG-3′ pattern [36,80]. Interestingly, *aflR* conserves the palindromic pattern 5′-TTAGGCCTAA-3′ within its own promoter, suggesting that, apart from being the major modulator of the AF gene cluster, *aflR* could also be auto-regulated [26,81]. *aflR* acts as a positive regulator of the AF gene cluster. In *A. flavus*, over-expression of *aflR* up-regulates several AF genes thereby increasing the production of aflatoxin 50-fold [82]. In the same way, *aflR* deletion in an *O*-methylsterigmatocystin (OMST)-accumulating strain of *A. parasiticus,* had a negative effect on aflatoxin pathway genes and hence on the production of OMST [83]. Recently, it has been demonstrated that different inhibitors of aflatoxin production are capable to down-regulate *aflR* expression [12,13,17,84,85,86]. Finally, it has been demonstrated that even if *aflR* is the main activator of the aflatoxin gene cluster, it interacts with *aflS* that plays the role of enhancer in the regulatory biosynthesis process [25]. 

### 4.2. The aflS Transcription Enhancer

*aflS* (previously named *aflJ*) was characterized by Meyers and co-workers [87] and was shown to be required for aflatoxin synthesis. In *A. parasiticus*, this gene encodes a 438-amino acid protein that has no important homology with any other proteins found in databases [22]. The *aflS* gene is located adjacent to *aflR* in the AF biosynthetic cluster, and *aflS* was also shown to be regulated by *aflR* [24]. Both genes share a 737-bp intergenic region from their translational starting sites [22]. In *A. flavus* and *A. parasiticus*, it was also demonstrated that *aflS* interacts with *aflR* but not with the biosynthetic enzymes, revealing the co-active function of *aflS* [25,88]. Disruption of *aflS* in *A. parasiticus* resulted in mutants with a 5- to 20-fold reduction in the expression of some AFB1 genes, such as *aflC*, *aflD*, *aflM* and *aflP,* and a subsequent loss in the ability to synthesize AF intermediates [25]. Otherwise, its overexpression in *A. flavus* resulted in higher levels of AFB1 production with 4- to 5-fold increase in the level of *aflC* and *aflD* expression as well as in averantin synthesis [88]. Even if it is still not clear how *aflS* increases the transcription levels of the genes involved in the AF pathway, it has been demonstrated that a dimer can be formed between both corresponding proteins in order to activate the AF gene cluster. This mechanism was proposed by Du et al. [88] who reported that this dimer-complex could recognize specific sites in the promoter regions of *aflC* and *aflD* (involved in early steps in the AFB1 biosynthesis) increasing their transcription and hence AFB1 production. Concerning the protein-dimer formation, it was further hypothesized that when both genes (*aflR* and *aflS*) are expressed normally, a protein complex made of 4 AflS for 1 AflR is formed, allowing correct binding to the promoter regions of the aflatoxin cluster genes [89]. Previous studies demonstrated that under certain AFB1 inhibitory conditions, a marked decrease in cluster gene expression and AFB1 production was observed with only low-level changes in *aflR* expression [90]. We obtained a similar result in one of our studies in which AFB1 levels were significantly reduced whereas *aflR* expression was not, although a major down-regulation of *aflS* and aflatoxin production was observed [15]. We suggested that the differential expression of *aflR* and *aflS* may have modified the ratio of available AflR and AflS proteins, leading to the formation of an insufficient number of complexes. Consequently, all the AflR-binding sites may not have been reached and the subsequent cluster transcription was not complete. The main argument supporting this hypothesis is the observation that genes encoding enzymes involved in the final stages of the AFB1 enzymatic cascade were more impacted than those involved in the initial steps. Therefore, the limited number of AflR/AflS complexes available might have been rapidly used at the beginning of AFB1 synthesis and were no longer available to properly activate the last cluster genes [18]. Even if great progress has been made in the characterization of *aflS*, the exact mechanism of action by which this gene modulates transcription of the aflatoxin pathway is still under investigation.

## 5. Global Regulation of Aflatoxin Production

Global regulatory factors here refer to genes that do not belong to the aflatoxin gene cluster but that have been demonstrated to have a link with aflatoxin production. In fact, aflatoxin biosynthesis can be modulated by different genes, external to the cluster, and that are themselves activated or repressed by environmental stimuli. Some of these influencing environmental stimuli are nutrient sources, exposure to environmental changes as well as oxidative fungal response. Many recent works demonstrated an interaction between the genes involved in the response to these environmental factors and the genes belonging to the aflatoxin cluster, even if the mechanisms of such interactions are often not completely understood. This is mainly due to the complexity of the pathways involved and their multiple interconnections. In this part of the review, to describe the recent data demonstrating links between global regulatory factors and AF production, genes interfering with aflatoxin production are classified in several groups based on their demonstrated function or role in cellular metabolism. The data are presented in summary tables including the genes involved, their known role in fungal metabolism and the works reporting their link with AF synthesis. When available, data obtained in *A. nidulans* and related to ST production are also included.

### 5.1. Growing Conditions

Growing conditions are one of the factors that have the most influence on the production of fungal secondary metabolites. They are highly dependent on many environmental variables. For instance, temperature and water activities are key parameters for fungal development and toxinogenesis [91,92] but to date the specific genes involved in such response and their link with aflatoxin cluster has not been described. However, it has been reported that several genes belonging to the aflatoxin cluster including the principal regulators *aflR* and *aflS* can be modulated by changes in temperature and water activity demonstrating a close link between these external factors and aflatoxin production [91]. Other environmental parameters including nutrient sources, pH and light exposure were found to interfere directly with aflatoxin production [31,93] and the role of some genes has been highlighted in such phenomenon.

#### 5.1.1. Nutrient Sources

##### Carbon Source

It has long been known that the availability and type of carbon source can modulate the number of secondary metabolites. Sugars are the favored carbohydrates for AFB1 production since they generate polyketide starter units (e.g., Acetyl-CoA) [94,95]. Simple sugars (e.g., sucrose, glucose, fructose, sorbitol) used as carbon sources have been correlated with higher levels of AF production in *A. flavus*, *A. parasiticus* and *A*. *nidulans* [29,96,97,98]. However, the use of D-glucal (a glucose-derivative not metabolized by fungi), used as the main source of sugar in the culture medium, inhibited AFB1 production [99]. 

Even if glucose is the best carbohydrate for the production of aflatoxins, they can be also produced using other carbon sources (e.g., ribose, xylose, or glycerol) [94]. In *Aspergillus*, the utilization of a given carbon source involves the sugar cluster. It contains four genes grouped in a 7.5 kB cluster located next to the AF gene cluster in *A. parasiticus* and *A. flavus* [96,100].

Carbon catabolic repression (CCR) is a strategic mechanism used by *Aspergilli* to preserve energy and adjust carbon catabolism in order to use the most favorable carbon source [93,101]. During this process the transcription factor (TF) CreA, along with the genes that interact with it, are presumed to be the main regulators (Table 1). The relation with AF production is first based on the fact that several genes belonging to the AF cluster have CreA-binding sites near their promoter regions [29]. In addition, it was demonstrated that this TF is needed for several fungal functions including AFB1 production in *A. flavus* [102].

##### Nitrogen Source

In *Aspergillus* species, nitrogen sources can affect aflatoxin and/or ST production in different ways and are regulated by the nitrogen metabolite repression mechanism [97]. AreA plays a crucial role in the process because it is responsible for modulating genes coding for the utilization of alternative sources of nitrogen. The nitrogen source is closely linked to AF production since some substrates (e.g., asparagine, ammonium salts, glutamate) support aflatoxin production while others do not (e.g., sodium nitrate, tryptophan) [22] (Table 2). 

#### 5.1.2. pH Conditions

pH is another extracellular condition to which fungal organisms react and that strongly modulates the production of secondary metabolites [115,116]. Indeed, the influence of pH has been linked to AF/ST production in *Aspergillus* species where *pac**C* seems to be the key transcription factor involved in the response to pH [117] (Table 3). In addition, the PacC signaling pathway includes six other proteins with Pal domain, involved in pH sensing [118]. 

#### 5.1.3. Light

Light stimulus is another important factor for fungi since it has a broad impact on fungal adaptation and survival. Light sensing can affect the circadian clock and growth, and has an impact on several morphological features and on the production of secondary metabolites [122] (Table 4). Light response in *Aspergilli* was shown to strongly involve the “velvet complex” where the global regulator *veA* governs a great number of genetic elements including photoreceptors [123,124,125]. 

### 5.2. Reproductive Processes

#### 5.2.1. Sexual Development

For years, it was believed that the reproductive process of *A. flavus* only occurred in an asexual way but then sexual reproduction was demonstrated in *A. flavus* and *A. parasiticus*. Both species were characterized as heterothallic species containing one of two mating-type genes: *MAT1-1* or *MAT1-2* (Table 5). In *A. flavus* sexual reproduction occurs within sclerotia when they recombine with the opposite mating type [142,143,144]. 

#### 5.2.2. Asexual Development

Secondary metabolite production is also coordinated with fungal development which, in turn, is intrinsically linked to conidiation (asexual reproduction) [31,93,97]. In fact, an interconnection between conidiation and AF/ST production has been demonstrated and several genes including *fadA*, *fluG*, *flb*- genes and *brlA,* were shown to be involved in this phenomenon [31,36] (Table 6). 

### 5.3. Oxidative Stress

In fungi, changes in environmental conditions can alter the normal intracellular balance between the production of reactive oxygen species (ROS) and the production of scavengers. In response to this phenomenon, several transcription factors are expressed to activate the enzymatic defenses that protect cells from excessive ROS levels and possible subsequent damage to DNA, proteins, and lipids [28]. In addition, their oxidative status has also been linked to the production of secondary metabolites and demonstrated to be a prerequisite for AFB1 production [45,164] (Table 7). In fact, it is proposed that, in *A. parasiticus* and *A. flavus,* aflatoxin production is part of the fungal oxidative stress response [67,164]. Indeed, AFs and their metabolic precursors (e.g., OMST, versicolorins, norsolorinic acid) are highly oxygenated molecules and thus subject to redox regulation [165]. AFB1 biosynthesis is activated by high levels of oxidative stress-inducing factors (e.g., lipid hydroperoxides) whereas several antioxidants (e.g., polyphenols) can block toxin production [166]. The relation between oxidative stress and secondary metabolites production has been thoroughly investigated in fungi. 

#### 5.3.1. Superoxide Dismutases and Catalases

Superoxide dismutases (SOD), catalases (CAT) and glutathione peroxidase (GPX) are key enzymes in fungal defense against ROS, thus mediating cellular defense against oxidative stress. SOD act as a first line of defense by converting the free radicals into hydrogen peroxide (H_2_O_2_) and oxygen (O_2_), then peroxidases and catalases convert H_2_O_2_ into H_2_O and/or O_2_ and H_2_O respectively [176]. Both also interact with AF production (Table 8).

#### 5.3.2. β-oxidation

β-oxidation of fatty acids is a fungal process that degrades fatty acids into acetate units [181]. Since AFB1 biosynthesis is triggered by acetate units, this pathway may be indirectly linked to AFB1 production [95,181] (Table 9). In addition, β-oxidation of fatty acids occurs in peroxisomes in *A. flavus* [182]. Vacuoles and vesicles are also participating in aflatoxin production since they are involved in final steps of aflatoxin synthesis and export [75,183]. 

### 5.4. Cell Signaling

Cell signaling in fungal cells helps overcome environmental stresses by activating rapid signal transduction systems throughout the cell that enable the organism to adapt to its surroundings [30]. G-protein coupled receptors (GPCR) and oxylipins are two of the actors involved in such processes. It has been shown that regulation of G-protein signaling plays an important role in secondary metabolite production [31,186] (Table 10) and that oxylipins influence secondary metabolism at a transcriptional level [187]. 

Many genes involved in different cell functions were found to be able to interact directly or indirectly with aflatoxin gene cluster and thus modulate the biosynthesis of these mycotoxins. In order to identify the possible connections between these genes and understand what could be the starting point of the modulation of aflatoxin production in response to diverse environmental stimuli, a schematic representation is proposed on Figure 4. 

## 6. Conclusions

As demonstrated, AFB1, one of the most hazardous mycotoxins contaminating foods and feeds, is produced by a very complex process involving numerous enzymes encoded by genes grouped in a cluster. The internal regulation of this process is also complex and under the direct dependence of two cluster-specific regulators (*aflR*-*aflS*) that interact together where it seems to be that the proportion of their corresponding proteins may be a key point in the activation of the gene cluster. The identification of natural products able to down modulate the expression of one of the two, without affecting fungal physiology could be a promising strategy to control aflatoxin B1 contamination of foods and feeds. It may allow avoiding toxin synthesis without any impact on biodiversity and no risk of resistance. Understanding how cluster-specific regulators are impacted by global environmental signals and the identification of genes involved is an ambitious goal. Fungal secondary metabolism is very diverse and complex and we are still at the beginning of its elucidation. The development of molecular tools will surely strongly help deciphering the role and function of numerous genes involved in secondary metabolite production. The development of mutant strains demonstrated the direct interaction of some genes primary involved in the response of environmental changes or adaptation and the aflatoxin cluster genes. These experiments were done using a targeted approach that brought useful information. It seems now important to identify the starting point of the interaction. The use of untargeted approaches such as RNAseq or micro-array techniques could help to define if there is the common function/gene that is implicated upstream the cluster in aflatoxin inhibition that can be observed after incubation of the toxigenic fungi with different inhibitors and/or in certain culture conditions. It could allow the identification of new targets to control aflatoxin production and ensure food safety using eco-friendly and sustainable means. However, to date, the still limited functional annotation of the genome of aflatoxin-producing fungal species remains a strong limit to this type of approach.

## Figures and Tables

**Figure 1 toxins-12-00150-f001:**
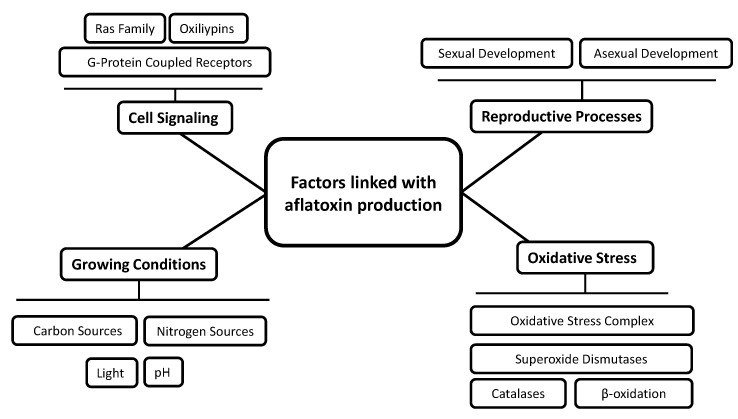
Schematization of the different factors linked with aflatoxin production.

**Figure 2 toxins-12-00150-f002:**
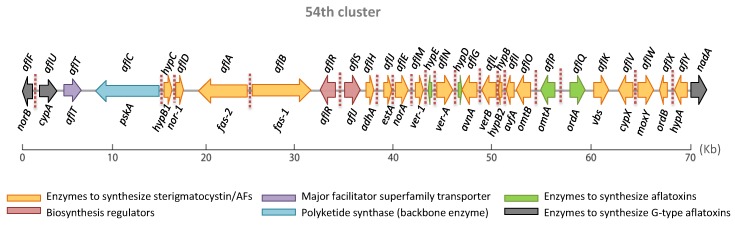
Organization of the aflatoxin gene cluster including the old and new cluster gene nomenclatures. This figure was adapted from the works of [24,33,37]. Red dotted lines represent the binding sites of AflR in the above pathway.

**Figure 3 toxins-12-00150-f003:**
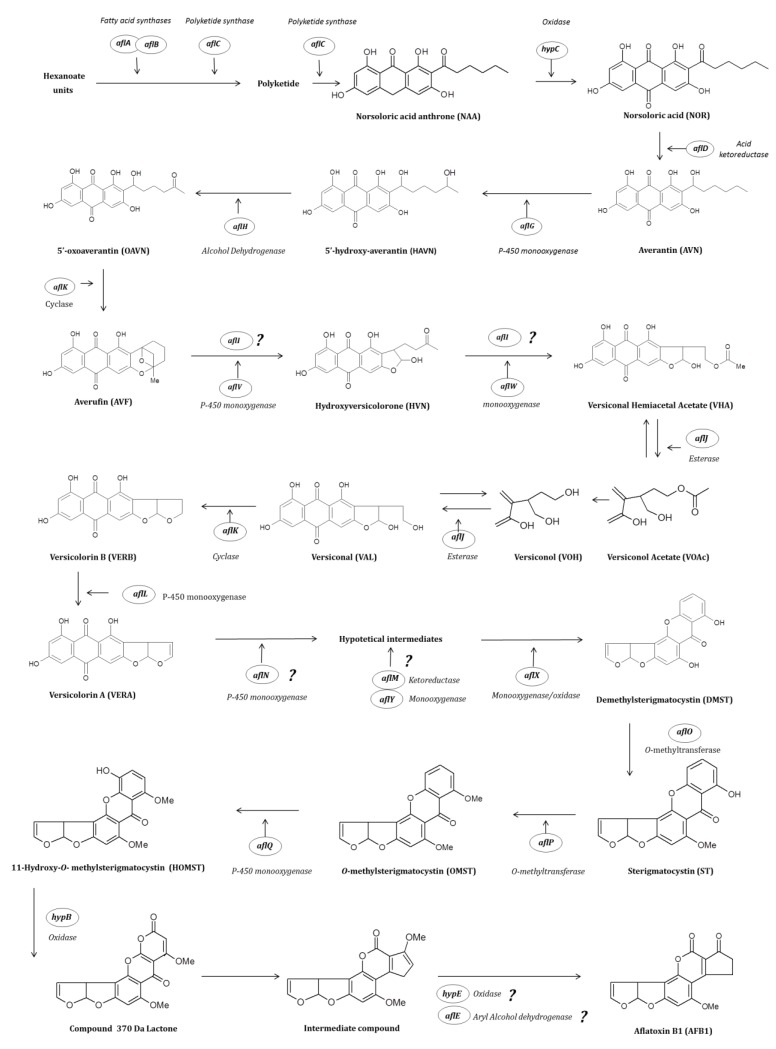
Schematization of the main intermediates produced during the AFB1 biosynthetic pathway and the confirmed or putative (indicated by?) level of intervention of the genes belonging to the AFB1 cluster. This figure is adapted from the works of [23,24,33,46,47,48,49,50,51,52,53,54,55].

**Figure 4 toxins-12-00150-f004:**
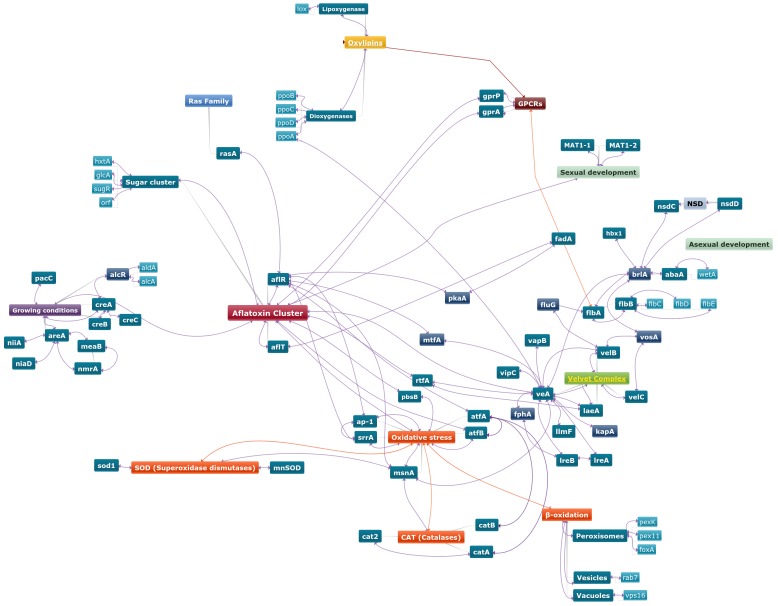
Schematic representation of a putative map of genes involved in aflatoxin regulation in diverse *Aspergillus* species.

**Table 1 toxins-12-00150-t001:** Genes involved in carbon catabolic repression.

Carbon Catabolic Repression	
Genes	Coding Function
*creA*	Cys_2_His_2_ Zinc finger transcription repressor
*creB*	Cys_2_His_2_ Zinc finger ubiquitin processing protease
*creC*	Cys_2_His_2_ Zinc finger ubiquitin processing protease
*alcR*	Regulon specific transcription factor
*aldA*	Aldehyde dehydrogenase
*alcA*	Alcohol dehydrogenase
Relation between genes or corresponding proteins in *Aspergilli* spp.:	
*creA* represses the expression of *aclR* while the latter is a positive regulatory factor for the genes *aldA* and *alcA* [101,103]. CreB-CreC form a protein complex that is essential for CreA function and stability [104].	
Demonstration of the connection with AF/ST synthesis	
Gene expression in the AF/ST gene cluster is regulated either positively or negatively by CreA depending on the fungal species [31,105]. **In** ***A. flavus:*** Δ*creA* strains produced insignificant amounts of aflatoxin in complete medium and their ability to colonize hosts was also impaired [102]. The repression of *creA* by gallic acid lead to the inhibition of AFB1 production [106]. **In** ***A. parasiticus:*** Several genes in the AFB1 cluster were demonstrated to have CreA-sites near their promoter regions [107].	

**Table 2 toxins-12-00150-t002:** Genes involved in nitrogen utilization.

Nitrogen Source	
Genes	Coding Function
*areA*	Zinc finger mediating nitrogen metabolite repression
*nmrA*	Repressive nitrogen
*meaB*	Regulatory protein
*niiA*	Nitrite reductase
*niaD*	Nitrate reductase
Relation between genes or corresponding proteins in *Aspergilli* spp.:	
*areA* and *meaB* are regulatory genes that mediate nitrogen metabolite repression [108,109] AreA regulates nitrate transporters binding at the intergenic regions of *niiA* and *niaD* [110] MeaB is a regulatory factor that activates NmrA, which is a repressor of AreA [109].	
Demonstration of the connection with AF/ST synthesis	
**In*****A. flavus***: *areA* was recently identified in *A. flavus* and demonstrated to be partially involved in aflatoxin production. The lack and overexpression of *areA* resulted in increased/decreased amounts of aflatoxin depending on the nitrogen source media [111]. The direct implication of AreA in *aflS* expression has been demonstrated in the presence of different sources of nitrate and ammonium [112]. Overexpressing *meaB* strains lost their capacity to produce AFs on seeds while Δ*meaB* produced only a few aflatoxins [109]. *niiA* is one of the genes that does not belong to the AFB1 gene cluster but that is presumed to be regulated by the major internal AF regulator *aflR* [26]. **In** ***A. parasiticus***: AreA binds to the GATA sites of the promoters of *aflR*/*aflS* [110,113]. *aflC* and *aflD* genes are expressed on ammonium and peptone media but not on nitrate sources [114]. **In** ***A. nidulans***: Nitrate medium increases the production of sterigmatocystin while ammonium medium does not [114].	

**Table 3 toxins-12-00150-t003:** Genes involved in pH modulation.

pH Impact	
Gene	Coding Function
*pacC*	Cys_2_His_2_ (C2-H2)- Zinc finger pH regulator
Relation between genes or corresponding proteins in *Aspergilli* spp.:	
The transcription factor PacC is strongly expressed under alkaline conditions [117].	
Demonstration of the connection with AF/ST synthesis	
**In*****A. flavus***: An increase in pH in nitrate-based medium results in lower levels of AF, while lower pH (4.0) resulted in a 10-fold increase in production of AF [119]. **In*****A. parasiticus:*** pH levels below 6.0 promote B-type aflatoxin production while higher levels induce G-type production [120]. A putative PacC binding site was identified in the *aflR* promoter region, but interestingly, alkaline conditions in which *pacC* is activated do not support AFB1 production [113]. *aflM* was higher expressed in acidic media than in neutral and alkaline media. Fungal growth reduced the pH of the medium and increased AF production with time [121]. **In** ***A. nidulans:*** Acidic conditions are more favorable for ST biosynthesis than neutral or alkaline ones [121]. *pacC*-mutant strains produced 10-fold less ST than the control. The expression level of *stcU* (*aflM* homologous) was lower in increased pH media [121].	

**Table 4 toxins-12-00150-t004:** Genes involved in the light response.

Light	
Genes	Coding function
*veA*	Global regulator
*vapB*	Methyltransferase
*vipC*	Methyltransferase
*fphA*	Phytochrome-like red light receptor
*kapA*	α transport carrier
*velB*	Velvet-like protein B
*laeA*	Putative methyltransferase
*velC*	Velvet-like protein C
*lreA*	Blue-light sensing protein
*lreB*	Blue-light sensing protein
*llmF*	LaeA-like methyltransferase
Relation between genes or corresponding proteins in *Aspergilli* spp.:	
Vea-LaeA-VelB form a trimeric complex called the velvet complex. VeA interacts with LaeA in the nucleus and with VelB in the cytoplasm and the nucleus. This trimeric complex, together with other light-receptor proteins, perceives light signals and is an essential coordinator of secondary metabolism and fungal development [124,125,126,127,128]. In dark conditions, the nuclear localization of VeA increases its interaction with LaeA to enhance production of secondary metabolites but also with VelB to induce sexual development [129]. FphA interacts with VeA, with LreB and LreA in the nucleus [123,130] and this protein complex is involved in red and blue light sensing [124]. KapA and VeA physically interact in dark conditions. KapA supports the entry of the VeA-VelB complex into the nucleus [126]. Both VelC and VelB form a protein dimer with VosA (involved in spore viability) [126]. VipC and VapB reduce the nuclear accumulation of VeA, thereby reducing secondary metabolism [131]. LlmF interacts with VeA controlling its subcellular location [129].	
Demonstration of the connection with AF/ST synthesis	
**In*****A. flavus:*** *veA* is essential for AFB1 production [132,133]. In null mutants of *veA* or *laeA*, no *aflR* expression was observed [134]. Δ*laeA* reduced *aflR*, *aflD* and *aflS* mRNA levels with no AF production [135]. Overexpression of *laeA* results in higher levels of AFB1 whereas Δ*laeA* strains do not affect aflatoxin production [109]. VeA governs 28 out of the 56 secondary metabolite gene clusters including the AF cluster [136]. **In** ***A. parasiticus:*** Deletion of *veA* resulted in the loss of the aflatoxin intermediate Versicolorin A. *VeA* is required for *aflR*/*aflS* expression [137]. **In** ***A. nidulans:*** Δ*veA* or Δ*veA/vipC* mutants were unable to produce sterigmatocystin but Δ*vipC* did not affect the mycotoxin production [131]. Both sexual development and sterigmatocystin production were repressed in light conditions [138]. Δ*veA* strains resulted in no *aflR* expression and neither ST production [139]. Deletion of the *laeA* gene inhibits *aflR* and *stcU* expression [140]. LreA, LreB and FphA modulate sterigmatocystin biosynthesis depending on light and on the presence of glucose [141]. FphA represses sexual development and ST production while LreA and LreB stimulate both processes. Blue light represses ST production while red light has the opposite effect [124]. LlmF is a negative factor for ST production [129].	

**Table 5 toxins-12-00150-t005:** Genes involved in sexual development.

Sexual Development	
Genes	Coding Function
*MAT1-1*	Mating type (alpha)
*MAT1-2*	Mating type (HMG)
Relation between genes or corresponding proteins in *Aspergilli* spp.:	
Either *MAT1-1* or *MAT1-2* is expressed in *A. flavus* and *A. parasiticus* strains and involved in sexual development [145].	
Demonstration of the connection with AF synthesis	
**In*****A. flavus:*** Strains of sexually developed *A. flavus* demonstrated that production of AF is highly heritable. In asexual development, non-aflatoxigenic populations are maintained while aflatoxigenicity increases in sexual development [146].	

**Table 6 toxins-12-00150-t006:** Genes involved in asexual development.

Asexual Development	
Genes	Coding Function
*fadA*	α-subunit of heterotrimeric G-protein
*fluG*	Developmental regulator
*brlA*	C_2_H_2_ zinc finger protein transcriptional activator of conidiophore
*abaA*	Transcription factor for conidia formation
*wetA*	Developmental regulatory protein
*nsdC*	Zinc-finger transcription factor
*nsdD*	Zinc-finger transcription factor
*pkaA*	Catalytic subunit of protein kinase A
*flbA*	RGS protein/developmental regulator
*flbB*	bZIP-type transcription factor
*flbC*	Putative C2H2 conidiation transcription factor
*flbD*	MYB family conidiophore development
*flbE*	Developmental regulator
*vosA*	Spore viability/Developmental regulator/Trehalose production
*rtfA*	RNA-pol II transcription elongation factor-like protein
*hbx1*	Homebox transcription factor
Relation between genes or corresponding proteins in *Aspergilli* spp.:	
*fluG* activates *flbA* which in turn represses *fadA* signaling [147]. *fadA* up-regulates *pkaA* [97]. *flbA* is a regulator of *flbB,* which regulates *flbC*, *flbD*, *flbE* (*flb* genes) [148]. *flb* genes are required with *fluG* for the correct expression of *brlA* [149]. *brlA* is a negative regulator of *abaA* and *abaA* is a repressor of *wetA* [126,150,151]. *brlA* is regulated by *veA* [139]. *vosA* is a repressor of *brlA* [152]. VosA also forms a protein-complex with VelB and VelC (velvet proteins) [126,127]. *nsdC* and *nsdD* are repressors of *brlA* [153].	
Demonstration of the connection with AF/ST synthesis	
**In*****A. flavus:*** Expression of *aflD*, *aflM*, and *aflP* is strongly reduced in *nsdC* deleted mutants. Loss of NsdC or NsdD resulted in developmental alterations that impact the ability of AflR to activate the expression of AF biosynthesis genes [153]. In fact, *A. flavus nsdC* mutants are unable to produce AF or any other secondary metabolites [154]. FadA governs both AF and ST biosynthesis [74,97]. Deleted strains of *rtfA* greatly reduced AFB1 biosynthesis but interestingly, *aflR, aflM and aflP* were up-regulated within the 3 first days of incubation and then down-regulated on the 4th and 5th day. Moreover, *rtfA* controls *veA* and *laeA* expression [155] and production of other secondary metabolites [156]. Disrupted *hbx1* strains produced null B-type aflatoxin, conidia and sclerotia [157]. Several genes belonging to the aflatoxin cluster were not expressed in *hbx1* mutant strains [158]. **In** ***A. parasiticus:*** Mutants defective in conidiation processes also had reduced levels of AF production [159]. FadA is presumed to regulate *aflT*, the MFS belonging to the AFB1 gene cluster [74]. **In** ***A. nidulans:*** *FadA* up-regulates *pkaA,* which down-regulates conidiation. PkaA also inhibits AflR activity by phosphorylation [77]. *fluG*-deleted strains lost their ability to produce sterigmatocystin [160]. Mutations in *flbA* and *fluG* blocked both sterigmatocystin production and sporulation [161]. Δ*fadA* and Δ*pkaA* mutants failed in normal processes of conidiation and sterigmatocystin biosynthesis [162]. Co-regulation of *brlA* and *aflR* by the *fadA* signaling pathway genes was reported [139]. Sterigmatocystin production was positively regulated by *rtfA* [163].	

**Table 7 toxins-12-00150-t007:** Genes involved in oxidative stress response.

Oxidative Stress Complex	
Genes	Coding Function
bZIP transcription factors	
*ap-1*	bZIP transcription factor
*atfA*	bZIP transcription factor
*atfB*	bZIP transcription factor
Stress Response Signaling Pathway	
*srrA*	Transcription factor
*msnA*	Transcription factor
*acyA*	Adenylate Cyclase
*pbsB*	MAP kinase kinase
Relation between genes or corresponding proteins in *Aspergilli* spp.:	
In cell systems, the *ap-1* gene is activated under both, pro-oxidant and antioxidant conditions [167]. Together, AtfB, SrrA, Ap-1, PbsB and MsnA form a regulatory network involved in oxidative stress response and secondary metabolite production [67,168]. AtfA may interact with AtfB in response to oxidative stress [67,169]. *acyA* was shown to regulate AF biosynthesis as well as to intervene in hyperosmotic and oxidative stress [170]. *pbsB* was reported to be involved in stress responses and AFB1 biosynthesis [171].	
Demonstration of the connection with AF/ST synthesis	
**In*****A. flavus:*** The deleted Δ*acyA* strains were unable to produce aflatoxin in contrast to the control and some genes in the AF gene cluster (*aflR* and *aflO*) were also down-regulated [170]. PbsB positively regulates AFB1 production through *aflR,* the major regulator of the AFB1 gene cluster, as well as other genes such as *aflC*, *aflD, aflK* and *aflQ* [171]. *msnA* deletion results in 50% more aflatoxins as well as higher levels of reactive oxygen species (ROS) [172]. *ap1* deletion reduces aflatoxin production while the expression of *aflM* and a*flP* was down-regulated even though *aflR* was up-regulated [173]. **In** ***A. parasiticus:*** AtfB binds to promoters of seven genes belonging to the AF gene cluster [174]. *ap-1* deletion increases AF production while the Ap-1 protein binds to the promoter region of the *aflR* gene [175].	

**Table 8 toxins-12-00150-t008:** Genes coding for fungal superoxide dismutases and catalases.

Superoxide Dismutases and Catalases	
Genes	Coding Function
*mnSOD*	Manganese superoxide dismutase
*sod1*	Cu, Zn superoxide dismutase
*catA*	Conidia-specific catalase
*catB*	Mycelial catalase
*hyr1*	Glutathione peroxidase
Demonstration of the connection with AF/ST synthesis	
**In*****A. flavus:*** *mnSOD* and the genes *aflA*, *aflM*, and *aflP* belonging to the AF gene cluster are co-regulated [67]. Deletion of *sod* reduced AF production [177]. Increased expression of the genes of *catA*, *cat2* and *sod1* as well as an increased CAT enzymatic activity were observed in the presence of the AF inhibitor piperine [15]. The same increase in CAT activity was also observed with another AF inhibitor, cinnamaldehyde [178]. **In** ***A. parasiticus:*** Inhibition of AF production induced by *Lentinula edodes* increases SOD enzymatic activity within its mechanism of action [179]. **In** ***A. nidulans***: Deletion of *mnSOD* increased both glutathione reductase and catalase activities while its overexpression reduced the activity of catalase but increased SOD activity [180].	

**Table 9 toxins-12-00150-t009:** Genes involved in β-oxidation.

β-oxidation	
Genes	Coding Function
*pexK*	Existence of peroxisome
*pex11*	Peroxisome proliferation
*foxA*	Regulation of fatty acid metabolism by *β* -oxidation
*rab7*	Vesicle marker
*vps16*	Vacuole marker
Relation between genes or corresponding proteins in *Aspergilli* spp.:	
In filamentous fungi, peroxisomes are crucial for primary metabolism and play a role in the formation of some secondary metabolites [182]. Aflatoxisomes (aflatoxin vesicles) are partially regulated by VeA [183]. *pex* mutants are able to grow on acetate medium but their growth is affected by fatty acids, indicating β-oxidation enzymes require a peroxisomal location [184].	
Demonstration of the connection with AF/ST synthesis	
**In*****A. flavus:*** An increase in the number of peroxisomes enhances AFB1 production [182]. **In*****A. parasiticus:*** An increase in the number of vesicles is positively correlated with AF accumulation/export [183]. Nor-1 (protein involved in the norsolorinic acid biosynthesis) mainly occurs in the cytoplasm and vacuoles [185].	

**Table 10 toxins-12-00150-t010:** Genes involved in cell signaling.

Cell Signaling	
Genes	Coding Function
*ppoA*	(oxylipin) Dioxygenase
*ppoB*	(oxylipin) Dioxygenase
*ppoC*	(oxylipin) Dioxygenase
*ppoD*	(oxylipin) Dioxygenase
*lox*	(oxylipin) Lipoxygenase
*gprK*	GPCR
*gprA*	GPCR
*gprP*	GPCR
*rasA*	GTP-binding protein
Relation between genes or corresponding proteins in *Aspergilli* spp.:	
GPCRs are involved in oxylipin response [30].	
Demonstration of the connection with AF/ST synthesis	
**In*****A. flavus:*** The GPCRs were shown to interact with AFB1 synthesis and its precursor, sterigmatocystin (ST). Deletion of *gprK* and *grpA* increased AF production compared to the control strain [30]. When all four *ppo* genes and the *lox* gene were disrupted simultaneously, the mutant strains showed reduced conidiation and increased AF production on maize and peanut seeds [169]. **In** ***A. nidulans:*** Δ*ppoA*; Δ*ppoB* and Δ*ppoC* mutants are unable to produce ST [187]. RasA has been demonstrated to control *aflR* activity [77]. GprH and GprM are negative regulators of ST biosynthesis [188].	

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
