# Peer review of "Aflatoxin Biosynthesis and Genetic Regulation: A Review"

_toxins, 2020, doi:10.3390/toxins12030150_

Round 1

Reviewer 1 Report

The authors reviewed the genetic regulation of aflatoxin B1 biosynthesis. There have been some published review articles on this topic (examples listed below). Therefore, this review article is of less significance and novelty, and would not of less interest to reads of Toxins.

Woloshuk, C. P., & Prieto, R. (1998). Genetic organization and function of the aflatoxin B1 biosynthetic genes. FEMS Microbiology Letters, 160(2), 169-176.

Yu, J., Chang, P. K., Ehrlich, K. C., Cary, J. W., Bhatnagar, D., Cleveland, T. E., ... & Bennett, J. W. (2004). Clustered pathway genes in aflatoxin biosynthesis. Appl. Environ. Microbiol., 70(3), 1253-1262.

Trail, F., Mahanti, N., & Linz, J. (1995). Molecular biology of aflatoxin biosynthesis. Microbiology, 141(4), 755-765.

Payne, G. A., & Brown, M. P. (1998). Genetics and physiology of aflatoxin biosynthesis. Annual Review of Phytopathology, 36(1), 329-362.

Do, J. H., & Choi, D. K. (2007). Aflatoxins: detection, toxicity, and biosynthesis. Biotechnology and Bioprocess Engineering, 12(6), 585-593.

Author Response

Indeed, we agree that the biosynthetic pathway of aflatoxins is well known and has been reviewed several times before. However, the recent development of molecular tools usable for fungal physiology investigation has led to many publications in last few years which allow a better understanding of the regulation of this pathway by global regulators and genes located outside of the cluster. In the present review, 98 references out of 189 (52%) were published after 2007 (year of publication of the most recent review listed) and 51 (27%) were published during the last 5 years. These papers mostly investigate the role of genes outside the aflatoxin cluster on the production of this toxin. However, to understand the levels of intervention of these genes, it is necessary to present again the biosynthetic pathway. So, in our opinion, the high number of recent publications justifies the interest of publishing an up to date review. Modifications were brought in the text to point out these recent data.

Reviewer 2 Report

The manuscript consist of an interesting review on the status of arts concerning the genetic regulation of Aflatoxin B1 (AFB1) production by Aspergillus spp.

The paper deals with a topic of relevant interest and supply an exhaustive description of the present knowledge about the AFB1 biosynthetic pathway together with its internal regulator mechanisms.

Moreover, the Authors efficiently highlights the function of external constraints that modulate the AFB1 biosynthesis. The review is comprehensive and  the literature data are reported and discussed in a clear way. I think it is a worthy paper with interesting information to report.

However, before publication, the following papers should be cited and discussed:

Line 242 The aflR transcription factor

Dhanamjayulu P et al. (2019). Inhibition of aflatoxin B1 biosynthesis and down regulation of aflR and aflB genes in presence of benzimidazole derivatives without impairing the growth of Aspergillus flavus. Toxicon. Dec;170:60-67. doi: 10.1016/j.toxicon.2019.09.018

Line 297 Role of External Factors Modulating AFB1 Production

Briefly, discuss the role of the following factors as environmental factors influencing AFB1 biosynthesis:

-temperature and water activity.

Gallo A et al. (2016). Effect of temperature and water activity on gene expression and aflatoxin biosynthesis in Aspergillus flavus on almond medium.Int J Food Microbiol. 217:162-9. doi: 10.1016/j.ijfoodmicro.2015.10.026.

Gizachew D et al. (2019). Aflatoxin B1 (AFB1) production by Aspergillus flavus and Aspergillus parasiticus on ground Nyjer seeds: The effect of water activity and temperature. Int J Food Microbiol. 296:8-13. doi: 10.1016/j.ijfoodmicro.2019.02.017.

-interactions with other microbes:

Nikbakht Nasrabadi E et al. (2013). Reduction of aflatoxin level in aflatoxin-induced rats by the activity of probiotic Lactobacillus casei strain Shirota. J Appl Microbiol. 114:1507-15. doi: 10.1111/jam.12148. Epub 2013 Feb 18.

Lappa IK, et al. (2019). Dual Transcriptional Profile of Aspergillus flavus during Co-Culture with Listeria monocytogenes and Aflatoxin B1 Production: A Pathogen-Pathogen Interaction. Pathogens. 2019 Oct 20;8(4). pii: E198. doi: 10.3390/pathogens8040198.

Line 337 Nitrogen source

Fasoyin OE et al. (2019). Regulation of Morphology, Aflatoxin Production, and Virulence of Aspergillus flavus by the Major Nitrogen Regulatory Gene areA. Toxins (Dc 10;11(12). pii: E718. doi: 10.3390/toxins11120718.

Author Response

We thank the reveiwer for his comments.

Some of the suggested references were added in the revised version of the manuscript:

Dhanamjayulu et al was added in lines 276-277.

Gall et al. and Gizachew et al. were added and briefly discussed lines 335-341

Fasoyin et al. was added in table 2 corresponding to "genes involved in nitrogen utilization" (line 377)

However, for Nikbakht et al. and Lappa et al., we agree with reviewer that interaction with other microbes may interfere with secondary metabolite production. However, in these papers, the genes involved upstream the aflatoxin cluster and that are involved in that interference are not investigated and we therefore think that these articles are difficult to include and discuss in the present review.

Reviewer 3 Report

It is an interesting review article. I suggest few minor corrections

Pag 1 row 30 endocrinal problems—correct is endocrine problems 

Pag 3 r. 83 and a oxidase= and an oxidase

Pag 5 r. 145 - aflK (vbs) was first associated= - aflK (vbs) was firstly associated

Pag 6 r.195 Conversion of SterigmatocystIn to Aflatoxin B1= Conversion of Sterigmatocystin into Aflatoxin B1

Pag 7 r. 252 and Pag 8 r. 254 palindrome motif= palindromic model (or pattern)

Pag 8 r  293 beginning= begining

Pag 11 Table 4, line 4 from the bottom Expression highly expressed in acidic media that in neutral = highly expressed in acidic media than in neutral

Author Response

We thank the reviewer for his comments and suggested corrections. All of them were included in the revised version and the whole text was reviewed and edited by a professional native English scientific editor (invoice attached).

Round 2

Reviewer 1 Report

N.A.